# Deep Generative Models for learning Coherent Latent Representations from Multi-Modal Data

## Abstract

The application of multi-modal generative models by means of a *Variational Auto Encoder* (VAE) is an upcoming research topic for sensor fusion and bi-directional modality exchange. This contribution gives insights into the learned joint latent representation and shows that expressiveness and coherence are decisive properties for multi-modal datasets. Furthermore, we propose a multi-modal VAE derived from the full joint marginal log-likelihood that is able to learn the most meaningful representation for ambiguous observations. Since the properties of multi-modal sensor setups are essential for our approach but hardly available, we also propose a technique to generate correlated datasets from uni-modal ones.

## 1 Introduction

*Auto Encoder* (AE), *Variational Auto Encoder* (VAE), and more recently *Disentangled Variational Auto Encoder* ($\beta$-VAE) have a considerable impact on the field of data-driven leaning of generative models. Furthermore, recent investigations have shown the fruitful applicability to *deep reinforcement learning* (DRL) as well as bi-directionally exchange of multi-modal data. VAEs tend to encode the data into latent space features that are (ideally) linearly separable as shown by Higgins et al. (2017a). They also allow the discovery of generative joint models (e.g. Suzuki et al. (2017)), as well as zero-shot domain transfer in DRL as shown by Higgins et al. (2017b).

However, a good generative model should not just generate good data and achieve a good quantitative score, but also gives a coherent and expressive latent space representation. This property is decisive for multi-modal approaches if the data shows correlation, as it is the case for every sensor setup designed for sensor fusion. With this contribution, we investigate the characteristic of the latent space as well as the quantitative features for existing multi-modal VAEs. Furthermore, we propose a novel approach to build and train a novel multi-modal VAE ($M^2$VAE) which comprises the complete marginal joint log-likelihood without simplifying assumptions. As our objective is the consideration of raw multi-modal sensor data, we also propose an approach to generate correlated multi-modal datasets from available uni-modal ones. Lastly, we draw connections to in-place sensor fusion and epistemic (ambiguity-resolving) active-sensing.

Section 2 comprises the related work on multi-modal VAEs. Our comprehensive approach (i.e. $M^2$VAE) is given in Sec. 3. Furthermore, we describe multi-modal datasets as well as the generation of correlated sets in Sec. 4 which are evaluated in Sec. 5. Finally, we conclude our work in Sec. 6.

## 2 Related Work

*Variational auto encoder* (VAE) combine neural networks with variational inference to allow unsupervised learning of complicated distributions according to the graphical model shown in Figure 1 (left). A $D_a$-dimensional observation $a$ is modeled in terms of a $D_z$-dimensional latent vector $z$ using a probabilistic decoder $p_{\theta_a}(z)$ with parameters $\theta$. To generate the corresponding embedding $z$ from observation $a$, a probabilistic encoder network with $q_{\phi_a}(z)$ is being provided which parametrizes the posterior distribution from which $z$ is sampled. The encoder and decoder, given by neural networks, are trained jointly to bring $a$ close to an $a'$ under the constraint that an approximate distribution needs to be close to a prior $p(z)$ and hence inference is basically learned during training.

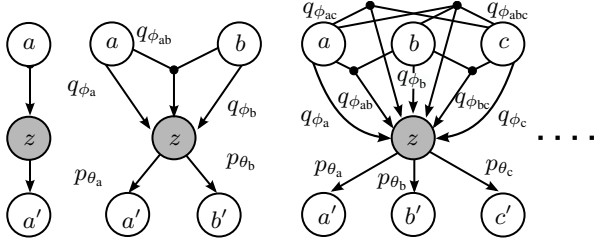

Figure 1: Evolution of full uni-, bi-, and tri-modal VAEs comprising all modality permutations

The specific objective of VAEs is the maximization of the marginal distribution $p(a) = \int p_\theta(a|z)p(z)\,\mathrm{d}a$. Because this distribution is intractable, the model is instead trained via *stochastic gradient variational Bayes* (SGVB) by maximizing the *evidence lower bound* (ELBO) $\mathcal{L}$ of the marginal log-likelihood $\log p(a) := L_a$ as

$$L_a \geq \mathcal{L} = \underbrace{-\,\mathrm{D}_{\mathrm{KL}}(q_\phi(z|a)\|p(z))}_{\text{Regularization}} + \underbrace{\mathbb{E}_{q_\phi(z|a)}\log(p_\theta(a|z))}_{\text{Reconstruction}}. \tag{1}$$

This approach proposed by Kingma & Welling (2013) is used in settings where only a single modality $a$ is present in order to find a latent encoding $z$ (c.f. Figure 1 (left)).

In the following chapters, we give a briefly comprise related work by means of multi-modal VAEs. Further, we stress the concept of two joint multi-modal approaches to derive the later proposed *variational Auto Encoder* (VAE).

## 2.1 MULTI-MODAL AUTO ENCODER

Given a set of modalities $\mathcal{M} = \{a,b,c,\dots\}$, multi-modal variants of *Variational Auto Encoder*s (VAE) have been applied to train generative models for multi-directional reconstruction (i.e. generation of missing data) or feature extraction. Variants are *conditional VAEs* (CVAE) and conditional multi-modal autoencoders (CMMA), with the lack in bi-directional reconstruction (Sohn et al. (2015); Pandey & Dukkipati (2017)). BiVCCA by Wang et al. (2016) trains two VAEs together with interacting inference networks to facilitate two-way reconstruction with the lack of directly modeling the joint distribution. Models, that are derived from the *variation of information* (VI) with the objective to estimate the joint distribution with the capabilities of multi-directional reconstruction were recently introduced by Suzuki et al. (2017). Vedantam et al. (2017) introduce another objective for the bi-modal VAE, which they call the triplet ELBO (tVAE). Furthermore, multi-modal stacked Auto Encoders (AE) are a variant of combining the latent spaces of various AEs ( Larochelle et al. (2007); Ranzato et al. (2006)) which can also be applied to the reconstruction of missing modalities (Ngiam et al. (2011); Cadena et al. (2016)). However, while Suzuki et al. (2017) and Vedantam et al. (2017) argue that training of the full multi-modal VAE is intractable, because of the $2^{|\mathcal{M}|}-1$ modality subsets of inference networks, we show that training the full joint model estimates the most expressive latent embeddings.

### 2.1.1 JOINT MULTI-MODAL VARIATIONAL AUTO ENCODER

When more than one modality is available, e.g. $a$ and $b$ as shown in Figure 1 (mid.), the derivation of the ELBO $\mathcal{L}_\mathrm{J}$ for a marginal joint log-likelihood $\log p(a) := L_\mathrm{J}$ is straight forward:

$$L_\mathrm{J} \geq \mathcal{L}_\mathrm{J} = \underbrace{-\,\mathrm{D}_{\mathrm{KL}}(q_{\phi_{ab}}(z|a,b)\|p(z))}_{\text{Regularization}} + \underbrace{\mathbb{E}_{q_{\phi_{ab}}(z|a,b)}\log(p_{\theta_a}(a|z))}_{\text{Reconstruction wrt. } a} + \underbrace{\mathbb{E}_{q_{\phi_{ab}}(z|a,b)}\log(p_{\theta_b}(b|z))}_{\text{Reconstruction wrt. } b} \tag{2}$$

However, it is not clear how to perform inference if the dataset consists of samples lacking from modalities (e.g. for samples $i$ and $k$: $(a_i,\varnothing)$ and $(\varnothing,b_k)$). Ngiam et al. (2011) propose training of a bimodal deep auto encoder using an augmented dataset with additional examples that have only a single-modality as input. We, therefore, name the resulting model of Eq. 2 *joint multi-modal VAE-Zero* (JMVAE-Zero).

### 2.1.2 JOINT MULTI-MODAL VARIATIONAL AUTO ENCODER FROM VARIATION OF INFORMATION

While the former approach cannot directly be applied to missing modalities, Suzuki et al. (2017) propose a *joint multi-modal VAE* (JMVAE) that is trained via two uni-modal encoders and a bi-modal en-/decoder which share one objective function derived from the *variation of information* (VI) of the marginal conditional log-likelihoods $\log p(a|b)p(b|a) =: L_{\mathrm{M}}$ by optimizing the ELBO $\mathcal{L}_{\mathrm{M}}$:

$$L_{\mathrm{M}} \geq \mathcal{L}_{\mathrm{M}} \geq \mathcal{L}_{\mathrm{J}} - \underbrace{\mathrm{D}_{\mathrm{KL}}(q_{\phi_{\mathrm{ab}}}(z|a,b)\|q_{\phi_{\mathrm{b}}}(z|b))}_{\text{Unimodal PDF fitting of encoder b}} - \underbrace{\mathrm{D}_{\mathrm{KL}}(q_{\phi_{\mathrm{ab}}}(z|a,b)\|q_{\phi_{\mathrm{a}}}(z|a))}_{\text{Unimodal PDF fitting of encoder a}} \tag{3}$$

Therefore, uni-modal encoders are trained, so that their distributions $q_{\phi_a}$ and $q_{\phi_b}$ are close to a multi-modal encoder $q_{\phi_{ab}}$ in order to build a coherent posterior distribution. The introduced regularization by Suzuki et al. (2017) puts learning pressure on the uni-modal encoders just by the distributions' shape, disregarding reconstruction capabilities and the prior $p(z)$. Furthermore, one can show that deriving the ELBO from the VI for a set of $\mathcal{M}$ observable modalities, always leads to an expression of the ELBO that allows only training of $\widetilde{\mathcal{M}} = \{m|m \in \mathcal{P}(\mathcal{M}), |m| = |\mathcal{M}| - 1\}$ modality combinations. This leads to the fact that for instance in a tri-modal setup, as shown in Fig. 1 (right), one can derive three bi-modal encoders from the VI, but no uni-modal ones.

## 3 MULTI-MODAL VARIATIONAL AUTO ENCODER APPROACH

While the objective of Wang et al. (2016), Ngiam et al. (2011), Suzuki et al. (2017), and Vedantam et al. (2017) is to exchange modalities bi-directionally (e.g. $a \to b'$), our primary concern is twofold: First, find a meaningful posterior distribution where the sampled statistics of an encoder network allows inference about further actions. Second, find an expression to jointly train all $2^{|\mathcal{M}|}-1$ permutations of modality encoders.

By successively applying logarithm and Bayes rules, we derive the ELBO for the multi-modal VAE (M²VAE) as follows: First, given the independent set of observable modalities $\mathcal{M} = \{a,b,c,\ldots\}$, its marginal log-likelihood $\log p(\mathcal{M}) =: L_{\mathrm{M}^2}$ is multiplied by the cardinality of the set as the neutral element $1 = |\mathcal{M}|/|\mathcal{M}|$. Second, applying logarithm multiplication rule, the nominator is written as the argument's exponent. Third, Bayes rule is applied to each term wrt. the remaining observable modalities to derive their conditionals. Further, we bootstrap the derivation technique in a bi- and tri-modal (c.f. tri-modal case in Sec. 6.1) case to illustrate the advantages. By excessively applying the scheme until convergence of the mathematical expression, it leads for a bi-modal set $\mathcal{M} = \{a,b\}$ to the following result:

$$L_{\mathrm{M}^2} = {}^2\!/\!{}_2 \log p(a,b) = {}^1\!/\!{}_2 \log p(a,b)^2 = {}^1\!/\!{}_2 \log p(a,b)p(a,b) = {}^1\!/\!{}_2 \log p(b)p(a|b)p(b|a)p(a) \tag{4}$$

$$= {}^1\!/\!{}_2(\log p(a) + \log p(b|a) + \log p(a|b) + \log p(b)) = {}^1\!/\!{}_2(L_{\mathrm{a}} + L_{\mathrm{M}} + L_{\mathrm{b}}) \tag{5}$$

This term can be written as inequality wrt. each ELBO of the marginals $L_{\mathrm{a}}$, $L_{\mathrm{b}}$ and conditionals $L_{\mathrm{M}}$:

$$2L_{\mathrm{M}^2} \geq 2\mathcal{L}_{\mathrm{M}^2} = \mathcal{L}_{\mathrm{a}} + \mathcal{L}_{\mathrm{b}} + \mathcal{L}_{\mathrm{M}} = \tag{6}$$

$$- \beta_{\mathrm{a}} \, \mathrm{D}_{\mathrm{KL}}(q_{\phi_{\mathrm{a}}}(z|a)\|p(z)) + \mathbb{E}_{q_{\phi_{\mathrm{a}}}(z|a)} \log(p_{\theta_{\mathrm{a}}}(a|z)) \tag{7}$$

$$- \beta_{\mathrm{b}} \, \mathrm{D}_{\mathrm{KL}}(q_{\phi_{\mathrm{b}}}(z|b)\|p(z)) + \mathbb{E}_{q_{\phi_{\mathrm{b}}}(z|b)} \log(p_{\theta_{\mathrm{b}}}(b|z)) \tag{8}$$

$$+ \mathbb{E}_{q_{\phi_{\mathrm{ab}}}(z|a,b)} \log(p_{\theta_{\mathrm{a}}}(a|z)) + \mathbb{E}_{q_{\phi_{\mathrm{ab}}}(z|a,b)} \log(p_{\theta_{\mathrm{b}}}(b|z)) - \beta_{\mathrm{ab}} \, \mathrm{D}_{\mathrm{KL}}(q_{\phi_{\mathrm{ab}}}(z|a,b)\|p(z)) \tag{9}$$

$$- \alpha \, \mathrm{D}_{\mathrm{KL}}(q_{\phi_{\mathrm{ab}}}(z|a,b)\|q_{\phi_{\mathrm{a}}}(z|a)) - \alpha \, \mathrm{D}_{\mathrm{KL}}(q_{\phi_{\mathrm{ab}}}(z|a,b)\|q_{\phi_{\mathrm{b}}}(z|b)). \tag{10}$$

Equation 6 is substituted by all formerly derived ELBO expressions lead to the combination of the uni-modal VAEs wrt. a and b (c.f. Eq. 7 to 8) and the JMVAE comprising the VAE wrt. the joint modality ab (c.f. Eq. 9) and mutual latent space (c.f. Eq. 10). Equation 7 and 8 have the effect that their regularizers care about the uni-modal distribution to deviate not too much from the common prior while their reconstruction term shapes the underlying embedding of the mutual latent space. We further apply the concept of $\beta$-VAE (Higgins et al. (2016; 2017a); Burgess et al. (2018)) to the regularizers via $\beta_*$ and adopt the factor $\alpha$ from Suzuki et al. (2017) for the mutual regularizer.

However, while $\beta$-VAE have the property to disentangle the latent space, our main concern is the balance between the input and the latent space using a constant normalized factor $\beta_{\text{norm}} = \beta_* D_*/D_z$.

If the derivation, which we leave out for the sake of brevity, is applied to the log-likelihood $L_{\text{M}^2{}_{\mathcal{M}}}$ of a set $\mathcal{M}$, one can show that it results into a recursive form consisting of JMVAEs' and M$^2$VAEs' log-likelihood terms

$$L_{\text{M}^2{}_{\mathcal{M}}} = \frac{1}{|\mathcal{M}|}\left(L_{\text{M}_{\mathcal{M}}} + \sum_{\widetilde{m}\in\widetilde{\mathcal{M}}} L_{\text{M}^2{}_{\widetilde{m}}}\right) \geq \frac{1}{|\mathcal{M}|}\left(\mathcal{L}_{\text{M}_{\mathcal{M}}} + \sum_{\widetilde{m}\in\widetilde{\mathcal{M}}} \mathcal{L}_{\text{M}^2{}_{\widetilde{m}}}\right) =: \mathcal{L}_{\text{M}^2{}_{\mathcal{M}}}. \tag{11}$$

While the derivation of Eq. 11 is given in Sec 6.1.3, the properties are as follows:

- the M$^2$VAE consist out of $2^{|\mathcal{M}|}-1$ encoders and $|\mathcal{M}|$ decoders comprising all modality combinations
- while it also allows the bi-directional exchange of modalities, it further allows the setup of arbitrary modality combinations having 1 to $|\mathcal{M}|$ modalities
- subsets of minor cardinality are weighted less and have a therefore minor impact in shaping the overall posterior distribution (vice versa, the major subsets dominate the shaping and the minor sets adapt to it)
- all encoder/decoder networks can jointly be trained using SGVB

## 4 DATA SETS

It is quite common in the multi-modal VAE community to model a bi-modal dataset as follows (Wang et al. (2016); Ngiam et al. (2011); Suzuki et al. (2017); Vedantam et al. (2017)): The first modality $a$ denotes the raw data and $b$ denotes the label (e.g. the digits' images and labels as one-hot vector wrt. the MNIST dataset). This is a rather artificial assumption and only sufficient when the objective is within a semi-supervised training framework. Real multi-modal data does not show this behavior as there are commonly multiple raw data inputs. Unfortunately, only complex multi-modal datasets of heterogeneous sensor setups exist (Ofli et al. (2013); Udacity (2016); Kragh et al. (2017)), which makes a comprehensive evaluation for VAEs futile. On the other hand, creating own multi-modal datasets is exhaustive since training generative models either demand dense sampling or supervised signals to form a consistent latent manifold (Bengio et al. (2012)).

While naïve consolidation of non-coherently datasets does not meet the conditions of data continuity, as discussed later, we propose a consolidation technique by sampling from superimposed latent spaces of various uni-modal trained CVAEs in Sec. 4.1. This approach allows the generation of multi-modal datasets from distinct and disconnected uni-modal sets. Second, we propose and bi-modal *mixture of Gaussians* (MoG) dataset to show particular behaviors of the various VAE approaches in Sec. 4.2.

### 4.1 MULTI-MODAL DATA GENERATION

Perry et al. (2010) state that Hebbian learning relies on the fact that the same objects are continuously transformed to their nearest neighbor in the observable space. Higgins et al. (2016) adopted this approach to their assumptions, that this notion can be generalized within the latent manifold learning. Further, neither a coherent manifold nor a proper factorization of the latent space can be trained if these assumptions are not fulfilled by the dataset. In summary, this means that observed data has to have the property of continues transformation wrt. to their properties (e.g. position and shape of an object), such that a small deviation of the observations results in proportional deviations in the latent space. We adopt this assumption for multi-modal datasets where observations should correlate if the same quantity is observed, such that a small deviation in the common latent representation between all modalities conducts a proportional impact in all observations. This becomes an actual fundamental requirement for any multi-modal dataset, as correlation and coherence are within the objective of multi-modal sensor fusion. In the following, we propose a technique to generate new multi-modal datasets, given different uni-modal enclosed sets which meet the former conditions.

A valuable property of the VAE's learned posterior distribution is, that it matches the desired prior quite sufficiently if only a single class is observed. This characteristic can be found again in the conditional VAE (CVAE) Kingma et al. (2014); Sohn et al. (2015) as it's training is supported by the ground truth labels of the observations. Thus, it actually builds non-related posterior distribution for each class label, where every distribution matches a given prior. Furthermore, we adopt the idea of $\beta$-VAE Higgins et al. (2017b) which learns disentangled and factorized latent representations. Combining the properties of both advantages allows the superimposing of latent manifolds from various uni-modal encoders as shown in Fig. 2 (Top-Right). Now, latent samples can be drawn from the posterior to operate all CVAE encoders, with the desired label, to generate continues multi-modal data.

To test the approach we consolidate MNIST (LeCun Yann et al. (1998)) and fashion-MNIST (Xiao et al. (2017)) to an *entangled-MNIST* (e-MNIST) set by sampling from the prior (i.e. $z \sim \mathcal{N}(0,\mathbf{I})$) to generate observation tuples from the corresponding encoder networks $p_{\theta_a}(a|z,C)$ and $p_{\theta_b}(b|z,C)$ with class label $C$. The network architecture is explained in Sec. 3. To avoid artifacts, only samples from within $2\sigma$ of the prior are obtained.

Furthermore, we train a bi-modal JMVAE on the newly generated data to depict properties of the different datasets. We are aware of the fact that consolidation of uni-modal datasets cannot be achieved easily since continuity is hardly measurable. Therefore, naïve consolidation results in a mixed dataset (i.e. mixed-MNIST) as shown in Fig. 2. To mimic this behavior and to achieve a fair comparison of the ELBO, we shuffle the generated fashion-MNIST per class label of e-MNIST to generate an equivalent *mixed-e-MNIST* (me-MNIST) set.

As shown in Fig. 2 (bottom), the JMVAE's latent space reveals that for m-MNIST single clusters share the same mean as the best representative of a single label, but the variance of any uni-modal trained encoder remains orthogonal. Thus, the continuity in the observations does not correlate with each other by any means. On the other hand, the e-MNIST set with continues samples shows the desired behavior of multi-modal datasets as the JMVAE trains a coherent distribution for all uni- and multi-modal encoders. These observations show that our proposed approach for generating new entangled datasets meet the formulated requirements of multi-modal datasets.

## 4.2 MoG-Example

We investigate a Mixture-of-Gaussians (MoG) distribution, as depicted in Fig. 3, as bi-modal observations to mimic the output of e.g. feature extractors or classifiers. While they commonly already provide linear separable observations, we focus on ambiguity resolving properties of the VAE in particular.

The bi-modal $(a,b)$ observations of Mixture-of-Gaussians have ten classes $(0,\ldots,9)$ each. $a$'s observations are organized on a grid where $(5,6,7)$ and $(0,8)$ result in ambiguous observations by sharing the same mean. $b$'s observations are organized on a circle where $(0,9)$ have ambiguous mean values.

This rather artificial experiment has the purpose to depict and evaluate ambiguous resolving properties of the VAEs. However, data of multi-modal sensor setups for complementary fusion show similar behavior, as various modalities are rectified to achieve a complete view of the scene (e.g. vision and grope to rectify objects). In that case, various dependencies of the generative process, given the class labels as factorized latent state representation $z = (z_0,\ldots,z_9)$, are possible. This is mimicked by the MoG-Experiment, in a simplified assumption, as: $p(a,b|z)$, $p(a|z_1,\ldots,z_4,z_9)$, and $p(b|z_1,\ldots,z_8)$.

## 5 Experiments

We apply the datasets explained in Sec. 4 to test and depict the capabilities of the M²VAE. First, we investigate the MoG data comprehensively. Second, we evaluate the ELBO of various approaches to the e-MNIST dataset. The VAEs are compared qualitatively, by visualizing the latent space, and quantitatively by performing lower bound tests $\mathcal{L}_{\widetilde{\mathcal{M}}}$ for every subset $\widetilde{\mathcal{M}} \subseteq \mathcal{M}$ wrt. to the decoding

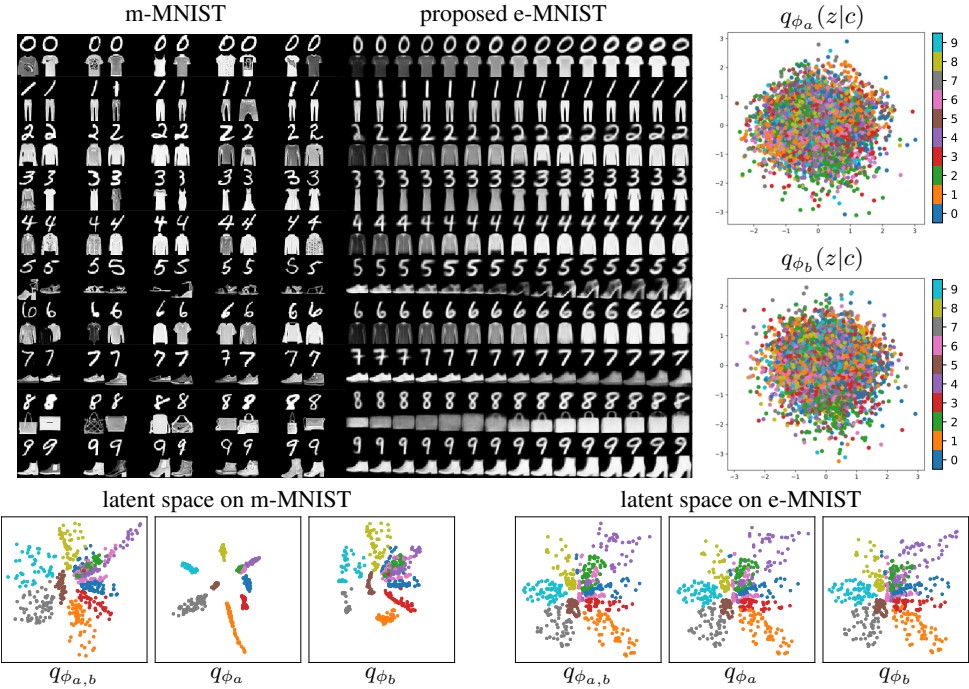

Figure 2: **Top-Left**: Depiction of naïve mixed MNIST (m-MNIST) vs. proposed entangled MNIST (e-MNIST). m-MNIST is pairwise plotted with the closest match of MNIST digits according to the mean-squared-error. The corresponding fashion-MNIST samples show no continuity nor correlation (despite the intended class correlation). e-MNIST shows the desired entanglement for changes of a single latent space factor. **Top-Right**: Latent space of the CVAE for the modalities $a$ (MNIST) and $b$ (fashion-MNIST). **Bottom**: Latent space of a trained JMVAE (c.f. Sec. 6.1.4). m-MNIST shows clear orthogonalization between modalities of the same class and segregation between classes (colorization is wrt. the CVAE legend). e-MNIST shows a coherently learned latent space between the uni- and multi-modal encoders. Thus, the JMVAE learns the correlation inside the dataset sufficiently ($\mathcal{L}_{a,b|\text{me-MNIST}} = -204.48$ vs. $\mathcal{L}_{a,b|\text{e-MNIST}} = -199.23$).

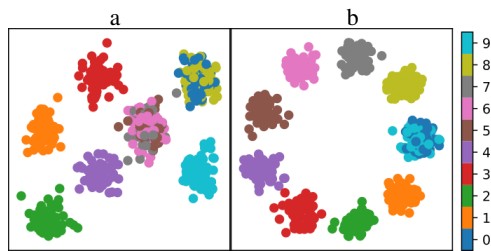

Figure 3: MoG input signals with for the modalities $a$ and $b$. The depicted observations are sampled for the corresponding modality for each class.

of all modalities $p_{\theta_\mathcal{M}}$:

$$\mathcal{L}_{\widetilde{\mathcal{M}}} = \mathbb{E}_{q_{\phi_{\widetilde{\mathcal{M}}}(z|\widetilde{\mathcal{M}})}} \log \frac{p_{\theta_\mathcal{M}}(\mathcal{M}|z)p(z)}{q_{\phi_{\widetilde{\mathcal{M}}}\left(z|\widetilde{\mathcal{M}}\right)} \tag{12}$$

with $p(z) = \mathcal{N}(z; \mathbf{0}, \mathbf{I})$. All VAE architectures can be found in Sec. 6.1.4.

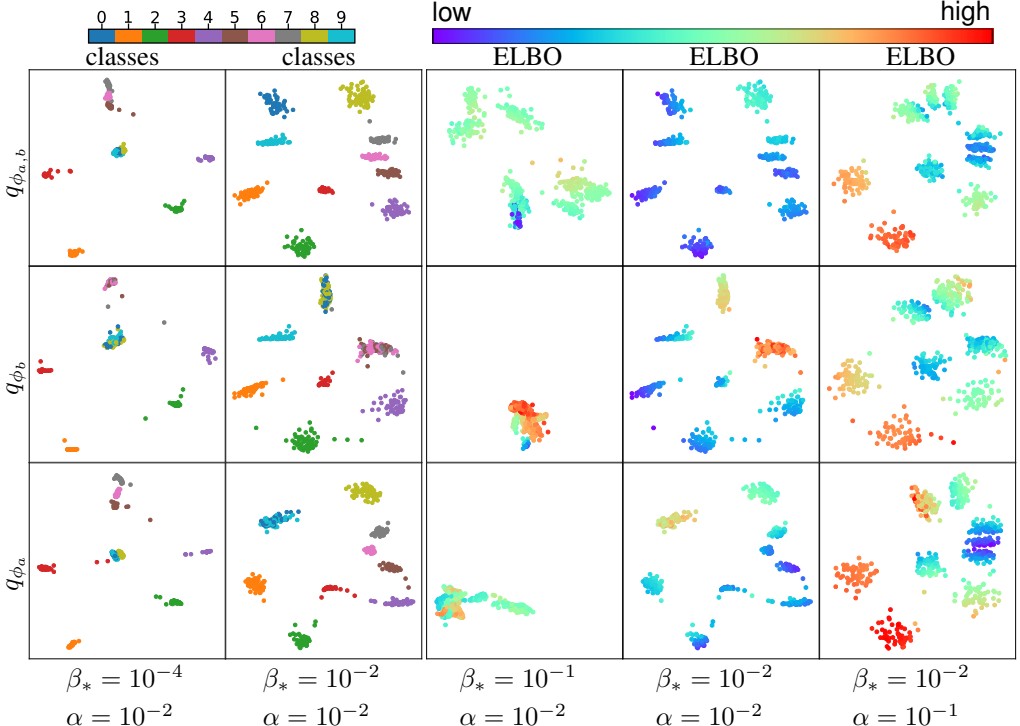

Figure 4: Latent space embeddings of the bi-modal MoG dataset by the three encoder networks of the $M^2VAE$. Classes and ELBO colorization is depicted for various parameter settings of $\beta_*$ and $\alpha$.

## 5.1 MoG-Experiment

We evaluate the latent space with the premise in mind, that a good generative model should not just generate good data but also gives a good latent representation $z$.

### 5.1.1 Parametrization

We first investigate the impact of the parameter set $(\beta_*, \alpha)$ on the $M^2VAE$ to find a latent space representation, which suits our needs to learn actions from it.

As the $\alpha$ parameter controls the mutual connection of all encoders in latent space, we found that a direct connection (i.e. $\alpha = 1.$) puts too much learning pressure on matching the mutual latent distributions between uni- and multi-modal encoders. Thus, classes which should be separated in the multi-modal latent space collapse to the mean distributions of the uni-modal encoders. For $\alpha \lesssim 10^{-2}$, the encoders are able to find an expressive latent space distribution by means of separable collapsed classes of uni-modal encoders, and expanded classes of multi-modal around it (c.f. Fig. 4 top/left).

By the findings of Higgins et al. (2017b), high $\beta$ values result in highly entangled factors in latent space whereas small normalized $\beta_{\mathrm{norm}} \lesssim 10^{-2}$ show pretty robust disentanglement in all their test cases. The impact of $\beta$ shows similar behavior on the $M^2VAE$ and thus, we chose small $\beta$ values of $\beta_{\mathrm{norm}} = 10^{-2}$ to relax the learning pressure caused by the prior. While the over optimization wrt. to the prior leads to a fuzzy generation of data $p(\mathcal{M}|z)$ and collapse in latent space, high relaxation ($\beta_{\mathrm{norm}} \ll 10^{-3}$) causes loss of expressiveness between uni- and multi-modal encoding of a single class by means of the difference in the ELBO.

It is worth noticing, that diverging $\beta$ parameters between multi- and uni-modal regularization (e.g. $\beta_{ab} \ll \beta_a$ or vice versa) results in lower ELBOs, but for the sake of expressiveness of latent embedding and the ELBO between encoders' embeddings. We argue that learning pressure should be applied equally to all encoders so that they experience a similar learning impact.

Another observation results from the fact, that the reconstruction loss of the M[2]VAE's objective causes learning of mean representatives of classes in the observation space. This causes the artifact, that if for instance three classes exist in the output space, where one represents the overall mean, and an uni-modal encoder only sees the collapse of classes to that particular mean value, the latent encoding of this uni-modal encoder will collapse to the same mean as well. However, while it is not longer separable (not even non-linearly) in latent space by its mean value, the ELBO for the observation drives up and gives, therefore, evidence about the embedding quality. This insight might be fruitful in terms of epistemic (ambiguity-resolving) tasks, where for instance an unsupervised reinforcement learning approach could use the ELBO as a signal to learn epistemic exploration.

### 5.1.2 COMPARISION

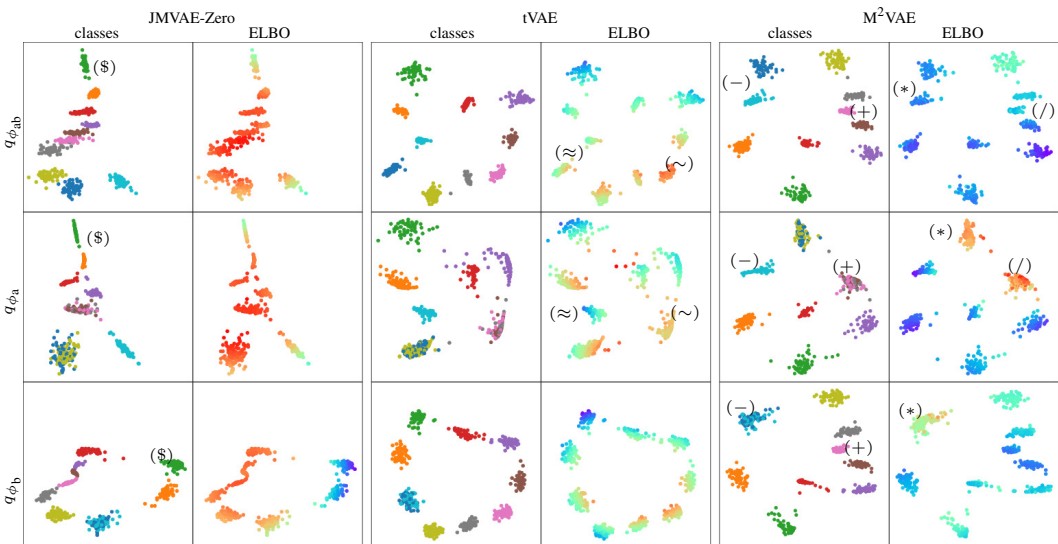

Figure 5: Bi-modal latent space embeddings by the three multi-modal VAEs JMVAE-Zero (left), tVAE (mid.), and M[2]VAE (right). The bi-modal input signals are an arrangement of the MoG distributions with ambiguities wrt. their mean values. The ELBO (colorization wrt. Figure 4) is estimated by Eq. 12 and is depicted qualitatively, as it can only be compared between encoders of the same approach.

Comparing the three approaches to estimate the multi-modal marginal log-likelihood by maximizing the ELBO, one can see from Fig. 5 that the most coherent latent space distribution was learned by the proposed M[2]VAE.

While the JMVAE-Zero learned similarities between $q_{\phi_{ab}}$ and $q_{\phi_a}$, it learned a complete new embedding for the classes $(1,2)$ with $q_{\phi_b}$ (denoted by $(\$)$). Furthermore, the ELBO per embedding allows no conclusion between the embeddings of the various encoders.

The tVAE founds a much more coherent embedding between the encoders. This was achieved by the fact, that first the full multi-modal VAE, consisting out of the encoder $q_{\phi_{ab}}$ and two decoder $p_{\theta_a}$ and $p_{\theta_b}$, was trained. Second, the decoder weights are pinned to train the remaining uni-modal networks which enforces coherence. However, the ELBO per embedding also does not allow any direct conclusion between the embeddings of the various encoders. This is depicted by $(\sim)$, where the multi-modal encoder $q_{\phi_{ab}}$ produces embeddings of higher energy than these of the uni-modal ones. This can happen as there is no regularizer which enforces the variational distribution of the encoders to match each other and thus, the KL-divergence may differ between the models for similar encodings.

The M[2]VAE, on the other hand, enforces the encoders inherently to approximate the same posterior distribution which can be seen by the strong coherence between all embeddings. Furthermore, classes which are separated in the multi-modal latent embedding collapse to the mean values in

the uni-modal ones as denoted by $(+)$ and $(-)$. This behavior is also rendered by the ELBO. As the $M^2$VAE makes ambiguous embeddings, the reconstruction loss drives up (c.f. $(*)$ and $(/)$).

The embeddings also show an interesting fact about the class $(0)$: As this class is only ambiguously detectable in the uni-modal case, all VAEs learn a linear separable and therefore unambiguous embedding if both modalities make an observation of this class (denoted by $(-)$ for the $M^2$VAE).

## 5.2 IN-PLACE SENSOR FUSION

Further, we introduce the concept of in-place sensor fusion using multi-modal VAEs. This approach is applicable in distributed active-sensing tasks where the latent space representation $z$ of observations $\mathcal{M}'$ (i.e. an object or point of interest was observed by a set of modalities) can be interpreted as inverse sensor model (c.f. Thrun et al. (2005)). This compressed information can be efficiently transmitted between all sensing agents and also be updated as follows: $z$ can be unfolded to the original observation using the VAE's decoder networks and combined with any new observation $m$ to update the information in-place $z \to z^*$ via

$$q_{\phi_{m \sqcup \mathcal{M}'}}(z^*|m, \mathcal{M}') \quad \text{with} \quad \mathcal{M}' = \bigcup_{m' in \mathcal{M}'} p_{\theta_{m'}}(m'|z). \tag{13}$$

However, a necessary requirement of Eq. 13 is that auto re-encoding (i.e. $z \to z$ via $q_{\phi_{\mathcal{M}'}}(z|\mathcal{M}')$) does not manipulate the information comprised by $z$ in an unrecoverable way (e.g. label-switching). Thus, we assume that VAEs tend to have a natural denoising characteristic (despite the explicit denoising Auto Encoders) which should re-encode any $z$ in a better version of its own by means of the reconstruction loss wrt. $z$. This behavior is shown in Fig. 6 where we underlay the latent representation with the reconstruction loss of every particular $z$. One can see the learned discrimination of the latent space by means of high entropy separating the clusters vicinity. Furthermore, initial $z$ values are auto re-encoded which draw the trajectories along their path in latent space. The observable properties of the VAE are that every seed converges to a fixed-point while performing descending steps on the latent space manifold. However, this statement is only valid in general for the proposed $M^2$VAE, as the JMVAE-Zero and tVAE learn no or only similar coherent latent spaces between the encoder networks. Thus, seeds may be attracted by wrong attractors which makes these approach not sufficient for in-place sensor fusion.

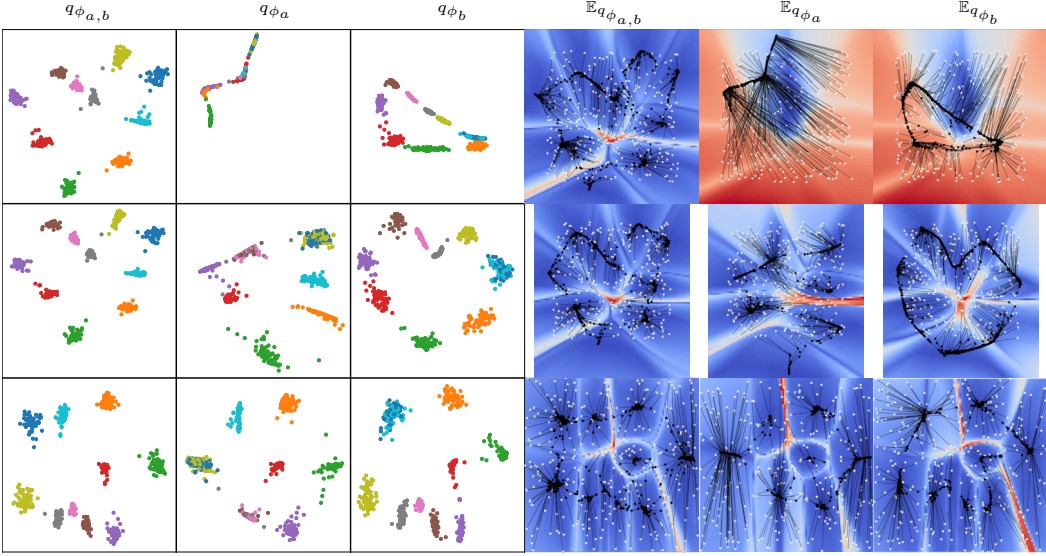

Figure 6: **From top to bottom**: JMVAE-Zero, tVAE, and $M^2$VAE. **Left**: Latent space representation with class colorization. **Right**: Corresponding colorization of the latent space for every $z$ obtained by auto re-encoding. White dots denote randomly drawn seeds which auto re-encoding steps are represented by the black trajectory. See Fig. 5 for legends.

### 5.3 E-MNIST EVALUATION

For this experiment, we estimated the ELBO by Eq. 12 to evaluate the performance of models JMVAE-Zero, tVAE, and $M^2$VAE. We chose the model wrt. to the evaluation in Fig. 4 with $\beta_{\text{norm}} = 0.01$ which is $\beta_* \approx 4$ for the given MNIST image dimension of $D_a = ||(28,28,1)||$ and $D_z = 2$. However, Tbl. 1 shows quantitatively and Fig. 7 depicts qualitatively that the pro-

Table 1: Evidence lower bound test for uni- and multi-modal setups of the VAEs (higher is better).

| $M^2$VAE | | | tVAE | | | JMVAE-Zero | | |
|---|---|---|---|---|---|---|---|---|
| $\mathcal{L}_{a,b}$ | $\mathcal{L}_a$ | $\mathcal{L}_b$ | $\mathcal{L}_{a,b}$ | $\mathcal{L}_a$ | $\mathcal{L}_b$ | $\mathcal{L}_{a,b}$ | $\mathcal{L}_a$ | $\mathcal{L}_b$ |
| $-10.75$ | $-10.91$ | $-16.01$ | $-23.6$ | $-101.28$ | $-88.75$ | $-24.19$ | $-131.05$ | $-99.71$ |

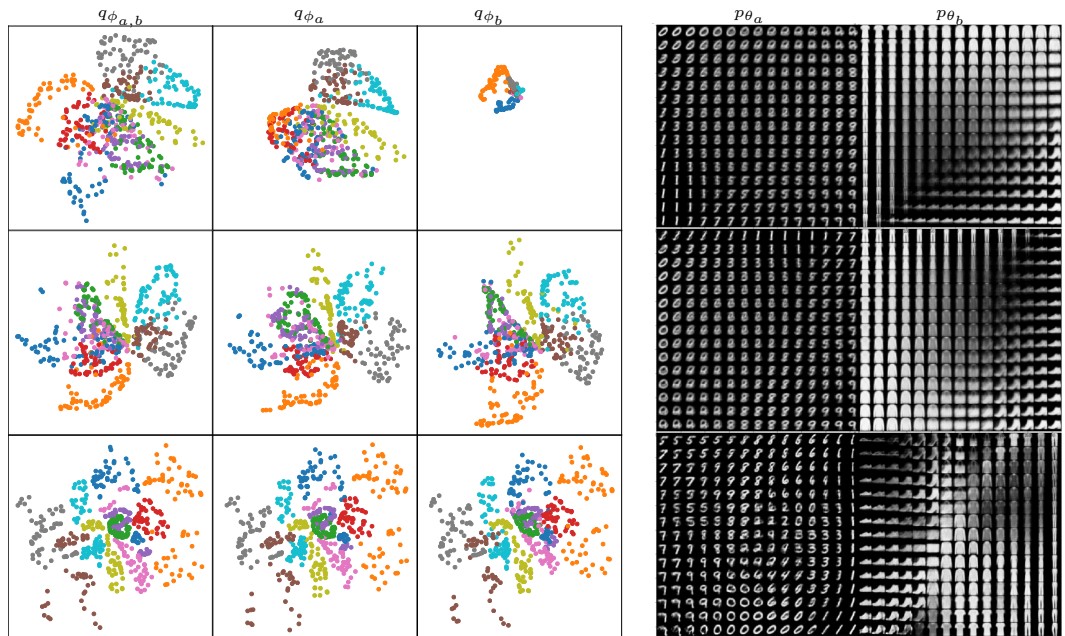

Figure 7: **From top to bottom**: JMVAE-Zero, tVAE, and $M^2$VAE. **Left**: Latent space representation with class colorization. **Right**: Reconstruction from latent space by applying the corresponding decoder networks. $z$ is sampled linearly within $2\sigma$ of the prior for all figures.

posed $M^2$VAE reaches the highest ELBO value, as well as it learns the most expressive latent space distribution. Furthermore, by sampling from the latent space for data generation, the $M^2$VAE reveals crisp reconstructions in comparison to the other approaches.

## 6 CONCLUSION

This work presents a novel multi-modal Variational Auto Encoder which is derived from the complete marginal joint log-likelihood. We showed that this expression can jointly be trained on an Mixture-of-Gaussian dataset with ambiguous observations, as well as on a complex dataset derived from MNIST and fashion-MNIST. Furthermore, we formulated requirements and characteristics for multi-modal data for sensor fusion and derived a technique to learn new datasets, namely the proposed entangled-MNIST, which suffice these requirements. Lastly, we developed the idea of in-place sensor fusion in distributed, active sensing scenarios and formulated the requirements, by means of auto re-encoding, to VAEs. This revealed the properties of VAEs, that they tend to denoise the observable data which leads to an attractor behavior in latent space. However, we performed all qualitative evaluations of the latent space with the premise in mind, that a good generative model

should not just generate good data but also gives a good latent representation. This does also correlate with the quantitative behaviors, as our proposed model achieved the highest ELBO values. Future work will concentrate on the integration of the ambiguous resolving characteristics to an epistemic-exploration scenario.

ACKNOWLEDGMENTS

This research was supported by 'CITEC' (EXC 277) at Bielefeld University and the Federal Ministry of Education and Research (57388272). The responsibility for the content of this publication lies with the author.

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

## APPENDIX

### 6.1 EXTENSION TO THREE MODALITIES

The proposed, as well as approach by Suzuki et al. (2017), can be extended to multiple modalities $\mathcal{M} = \{a, b, c\}$. The conditional marginal log-likelihood of $a$ can be written as

$$\log p(a|b,c) = \mathcal{L}_{\widetilde{M}_a} + D_{KL}(q(z|\mathcal{M})\|p(z|\mathcal{M})) \geq \mathcal{L}_{\widetilde{M}_a}. \tag{14}$$

#### 6.1.1 JMVAE FOR THREE MODALITIES

The VI between a set of distributions $\mathcal{M}$ can be written as $-\mathbb{E}_{p(\mathcal{M})} \sum_{m \in \mathcal{M}} \log p(m|\mathcal{M} \setminus m)$, which leads to an expression of maximizing the ELBO of negative VI (c.f. Suzuki et al. (2017)). Following this approach, the log-likelihood $L_{3M}$ can be expressed by the ELBOs, by utilizing Eq. 14, of their conditionals and KL divergence:

$$L_{3M} = \log p(a|b,c) + \log(p(b|a,c)) + \log(p(c|b,c)) \tag{15}$$

$$\geq \mathcal{L}_{\widetilde{M}_a} + \mathcal{L}_{\widetilde{M}_b} + \mathcal{L}_{\widetilde{M}_c} \tag{16}$$

$$\geq \mathcal{L}_{\widetilde{J}} - D_{KL}(q(z|a,b,c)\|p(z|b,c)) \tag{17}$$

$$- D_{KL}(q(z|a,b,c)\|p(z|a,c)) - D_{KL}(q(z|a,b,c)\|p(z|b,c)) \tag{18}$$

with $\mathcal{L}_{\widetilde{J}}$ being the joint ELBO of a joint probability $p(\mathcal{M})$ which expression is analog to Eq. 2.

### 6.1.2 M²VAE FOR THREE MODALITIES

Applying the proposed scheme to the joint log-likelihood of three modalities results in the following expression:

$$L_{3\text{M}^2} \tag{19}$$
$$= {}^3/_3 \log p(a,b,c) = {}^1/_3 \log p(a,b,c)^3 \tag{20}$$
$$= {}^1/_3 \log p(a,b,c)p(a,b,c)p(a,b,c) \tag{21}$$
$$= {}^1/_3 \log p(a,b)p(b,c)p(a,c)p(a|b,c)p(b|a,c)p(c|a,b) \tag{22}$$
$$= {}^1/_3(\log(p(a,b)) + \log(p(b,c)) + \log(p(a,c)) \tag{23}$$
$$\quad + \log p(a|b,c) + \log p(b|a,c) + \log p(c|a,b)) \tag{24}$$
$$= {}^1/_3({}^2/_2(\log p(a,b) + \log p(b,c) + \log p(a,c)) + L_{3\text{M}}) \tag{25}$$
$$= {}^1/_6\left(\log p(a,b)^2 + \log p(b,c)^2 + \log p(a,c)^2\right) + {}^{L_{3\text{M}}}/_3 \tag{26}$$
$$= {}^1/_6\left(L_{\text{M}^2{}_{ab}} + L_{\text{M}^2{}_{bc}} + L_{\text{M}^2{}_{ac}}\right) + {}^1/_3 L_{3\text{M}} \tag{27}$$

From here on, one can substitute all log-likelihoods given the expressions in Sec. 3 and **??**, to derive the ELBO $\mathcal{L}_{3\text{M}^2}$.

### 6.1.3 M²VAE DERIVATION

$$L_{\text{M}^2{}_\mathcal{M}} = \log p(\mathcal{M}) \overset{\text{mul. 1}}{=} {}^{|\mathcal{M}|}/_{|\mathcal{M}|} \log p(\mathcal{M}) \overset{\text{log. mul.}}{=} {}^1/_{|\mathcal{M}|} \log p(\mathcal{M})^{|\mathcal{M}|} \tag{28}$$

$$\overset{\text{Bayes}}{=} {}^1/_{|\mathcal{M}|} \sum_{m \in \mathcal{M}} \log p(\mathcal{M} \setminus m)p(m|\mathcal{M} \setminus m) \tag{29}$$

$$\overset{\text{log. add}}{=} {}^1/_{|\mathcal{M}|} \sum_{m \in \mathcal{M}} \log p(\mathcal{M} \setminus m) + \log p(m|\mathcal{M} \setminus m) \tag{30}$$

The expression $\sum_{m \in \mathcal{M}} \log p(m|\mathcal{M} \setminus m)$ is the general form of the marginal log-likelihood for the *variation of information* (VI), as introduced by Suzuki et al. (2017) for the JMVAE, for any set $\mathcal{M}$. Thus, it can be directly substituted with $L_{\text{M}_\mathcal{M}}$. The expression $\sum_{m \in \mathcal{M}} \log p(\mathcal{M} \setminus m)$ is the combination of all joint log-likelihoods of the subsets of $\mathcal{M}$ which have one less element. Therefore, this term can be rewritten as

$$\sum_{m \in \mathcal{M}} \log p(\mathcal{M} \setminus m) = \sum_{\widetilde{m} \in \widetilde{\mathcal{M}}} \log p(\widetilde{m}) \tag{31}$$

with $\widetilde{\mathcal{M}} = \{m | m \in \mathcal{P}(\mathcal{M}), |m| = |\mathcal{M}| - 1\}$ Finally, $\log p(\widetilde{m})$ can be substituted by $L_{\text{M}^2{}_{\widetilde{m}}}$ without loss of generality. However, it is worth noticing that substitution stops at the end of recursion and therefore, all final expressions $\log p(\widetilde{m}) \; \forall \; |\widetilde{m}| \equiv 1$ remain. $\square$

### 6.1.4 NETWORK ARCHITECTURE

We designed all VAEs such that the latent space prior is given by a Gaussian with unit variance. Furthermore, all VAEs sample from a Gaussian variational distribution that is parametrized by the encoder networks. A summary of all architectures used in this paper can be seen in Tbl. 2. The reconstruction loss for calculating the evidence lower bound was performed by *binary cross-entropy* (BCE) for the e-MNIST and *root-mean-squared error* (RMS) for the MoG experiment.

Furthermore, the CVAE for training the e-MNIST dataset is designed as depicted in Tbl. 3.

Table 2: Various VAE architectures and optimizers for the e-MNIST and MoG experiments. um/mm stand for uni- and multi-modal while fc refers to fully-connected layers.

| Issue | VAE | Optimizer | | VAE architecture |
|---|---|---|---|---|
| e-MNIST | JMVAE-Z. | adam | encoder | fc 2x784-2x128-2x64-concat-64-2 (ReLU) |
| | | | decoder | fc 2x64-2x128-2x786 (tanh) |
| e-MNIST | tVAE | adam | um enc. | fc 784-128-64-2 (ReLU) |
| | | | mm enc. | fc 2x784-2x128-2x64-concat-64-2 (ReLU) |
| | | | decoder | fc 2x64-2x128-2x786 (tanh) |
| e-MNIST | M²VAE | adam | um enc. | fc 784-128-64-2 (ReLU) |
| | | | mm enc. | fc 2x784-2x128-2x64-concat-64-2 (ReLU) |
| | | | decoder | fc 2x64-2x128-2x786 (tanh) |
| MoG | JMVAE-Z. | rmsprop | encoder | fc 2x2-2x128-concat-64-2 (ReLU) |
| | | | decoder | fc 2x128-2x2 (tanh) |
| MoG | tVAE | rmsprop | um enc. | fc 2x2-2x128-2x2 (ReLU) |
| | | | mm enc. | fc 2x2-2x128-concat-64-2 (ReLU) |
| | | | decoder | fc 2x128-2x2 (tanh) |
| MoG | M²VAE | rmsprop | um enc. | fc 2-128-2 (ReLU) |
| | | | mm enc. | fc 2x2-2x128-concat-64-2 (ReLU) |
| | | | decoder | fc 2x128-2x2 (tanh) |

Table 3: CVAE architecture for each dataset MNIST and fashion-MNIST. The label as one-hot-vector is concatenated after the convolution layers and fed into the fully-connected (fc) layers. For convolutional architectures the numbers in parenthesis indicate strides, while padding is always *same*.

| CVAE architecture | |
|---|---|
| encoder | conv 1x2x2-64x2x2 (2)-64x3x3-64x3x3-concat label C-fc 128-2 |
| decoder | concat label C-fc 128-deconv reverse of encoder (ReLU) |

