# OpenReview forum: "Deep Generative Models for learning Coherent Latent Representations from Multi-Modal Data"
_ICLR.cc/2019/Conference_

### Official Review · AnonReviewer2 · 2018-11-01
**Unclear about the novelty of the objective. Clarity could be improved.**

**Rating:** 4
**Confidence:** 4

**Review:**


This paper proposes an objective, M^2VAE, for multi-modal VAEs, which is supposed to learn a more meaningful latent space representation. To summarize my understanding of the proposed objective, in the bi-modal case, it combines both objectives of TELBO [1] and JMVAE-kl [2] with some hyperparameters to learn the uni-modal encoders. The terms of Eqns 7,8, and 9 are equivalent to TELBO and Eqns 9 and 10 are JMVAE-kl. It would be very beneficial for the readers if you could more clearly contrast your objective with the related work given how similar they are.

Given these similarities between objectives, its unclear why JMVAE-Zero was chosen over JMVAE-kl as a baseline. Furthermore, the reasoning for the improvement of the ELBO of M^2VAE over the baselines in Section 5.3 is unclear, given the similarities between the objectives.

The qualitative figures throughout the paper are hard to interpret. By looking at Fig 4., I cannot tell which latent space is best.
“one can see from Fig. 4 that the most coherent latent space distribution was learned by the proposed M^2VAE”
What is meant by ‘coherent latent space’?

This paper was hard to follow and there are a number of typos throughout the paper. For instance, the labels within Fig 4 and the caption contradict themselves. If the clarity and quality of the writing could be improved then perhaps the contributions may become more evident.

[1] R. Vedantam, I. Fischer, J. Huang, and K. Murphy. Generative Models of Visually GroundedImagination. ArXiv e-prints, May 2017.
[2] M. Suzuki, K. Nakayama, and Y. Matsuo. Improving Bi-directional Generation betweenDifferent Modalities with Variational Autoencoders. ArXiv e-prints, January 2018

---

> ### Author Response · Authors · 2018-11-08
> **Add takeaway message and JMVAE-Zero vs. JMVAE-kl**
>
> We’d like to thank the reviewer for their thorough review and helpful suggestions. We will improve our work further but also want to start a discussion to clarify the significance of our contribution.
>
> From our point of view, current VAE approaches do not explicitly care or investigate the coherence of the latent space in multi-modal cases. Thus, the encoder networks for example in a bi-modal case, i.e. q_\phi_a(z_a|a), q_\phi_b(z_b|b), and q_\phi_{a,b}(z_{a,b}|a,b), may project the observations into different latent spaces (z_a != z_b != z_{a,b}). This happens for the tVAE and JMVAE-Zero, as qualitatively shown in our results, due to the fact that these approaches do not maintain a fully analytical way for deriving the VAE’s objective from the joint marginal log-likelihood (JMLL). In our work, the VAE’s (namely M²VAE) objective has been consequently derived from the JMLL and therefore also disproves statements in related papers, which state that this approach is not applicable. As a result, our model shows coherence in the latent spaces (z_a = z_b = z_{a,b}) regarding the sampled latent space as well as the ELBO for each sample which is, from our point of view, a significant and unique finding in the field of multi-modal generative models. Furthermore, the trained latent space should serve as a much better representation, i.e. feature extractor for subsequent models, which is out of scope for this contribution.
>
> Furthermore, we choose the JMVAE-Zero over JMVAE-kl for the following reasons: JMVAE-Zero is a special case of the JMVAE-kl which is equivalent of using the ELBO derived from the joint marginal log-likelihood (JMLL) p(a,b). In general, the JMVAE-kl derives the ELBO from the variation of information log(p(a|b))+log(p(b|a)). For the sake of consistency, we chose only approaches which derive the objective from the JMLL.
>
> We apologize for the possibly unstructured Figure 4 and try to rephrase the takeaway message as follows: If the latent spaces (z_a, z_b, z_{a,b}) found by the various encoders (q_\phi_a(z_a|a), q_\phi_b(z_b|b), and q_\phi_{a,b}(z_{a,b}|a,b)) are the same, z_a and z_b should be sub-spaces of z_{a,b}. This should result in a congruence of samples and in a similarity of the ELBO per sample between the subspaces. These statements only hold for our proposed M²VAE as shown in the numerous examples in Figure 4. However, we will de-clutter Figure 4 and upload a new version with this answer.

---

> > ### Comment · AnonReviewer2 · 2018-11-15
> > **Response**
> >
> > "in a bi-modal case, i.e. q_\phi_a(z_a|a), q_\phi_b(z_b|b), and q_\phi_{a,b}(z_{a,b}|a,b), may project the observations into different latent spaces (z_a != z_b != z_{a,b})"
> >
> > The fact that z_a != z_b != z_{a,b} should be expected if a and b provide different information. I don't see the problem with this.
> >
> > "This happens for the tVAE and JMVAE-Zero, as qualitatively shown in our results, due to the fact that these approaches do not maintain a fully analytical way for deriving the VAE’s objective from the joint marginal log-likelihood (JMLL)"
> >
> > The model learned by tVAE is derived from the JMLL. q(z|a) and q(z|b) are learned separately from the model.
> >
> > "For the sake of consistency, we chose only approaches which derive the objective from the JMLL."
> >
> > Didn't you say that tVAE and JMVAE-Zero were not derived from the JMLL, yet you compared to them? So you could also compare to JMVAE-kl given how similar it is to your model. Right?

---

> > > ### Author Response · Authors · 2018-11-16
> > > **Answers to the comments**
> > >
> > > We thank the reviewer for the discussion and want to explain the following concerns as follows:
> > >
> > > First of all, we like to give some definitions to avoid confusion:
> > > A: Is the first dataset (e.g., MNIST) from which we can draw a sample "a" (e.g., a picture showing the letter "0")
> > > B: Is the second dataset (e.g., fashion-MNIST) from which we can draw a sample "b" (e.g., a picture showing a "t-shirt")
> > > AB: Is the entangled dataset (c.f. Figure 2 "proposed e-MNIST") from which we can draw a joint sample "a,b"
> > > z_a: Is the projection of sample "a" using q_{\phi_a} into the latent space
> > > z_b: vice versa
> > > z_{a,b}: Is the projection of the joint sample "a,b" using q_{\phi_{a,b}} into the latent space
> > >
> > > ----------------------------------------------------------------------------------------------------------------------------------------
> > >
> > > ----------------- Comment by the reviewer -----------------
> > > "The fact that z_a != z_b != z_{a,b} should be expected if a and b provide different information. I don't see the problem with this."
> > >
> > > ----------------- Answer by the authors -----------------
> > > This statement is absolutely correct and holds if two datasets A and B don't show any correlation. This would also mean that both datasets have a different generative model. On the other hand, if the datasets share some information (for instants in the m-MNIST in Figure 2 where "0"s occur together with "t-shirt"s, "7"s together with "heel"s, etc.) they might also share the same generative model and therefore come frome the same latent representation.
> > >
> > > Referring to Figure 7, we trained q_{\phi_{a}}, q_{\phi_{b}}, and q_{\phi_{a,b}} for the JMVAE-Zero, tVAE, and the proposed M²VAE on the AB dataset. Since we ensure a strong correlation not only per label but also per sample in the e-MNIST dataset, all q_{\phi_*} should project any sample "a", "b", or "a,b" drawn from AB onto the same latent representation, resulting in z_a = z_b = z_{a,b}. If this is not the case, the VAE was not able to find the common generative model.
> > >
> > > A disjoint latent space example can be seen in the latent space of the JMVAE-Zero (top of Figure 7), where the projections of q_{\phi_{a}} and q_{\phi_{b}} are entirely disjoint. Furthermore, the projections from q_{\phi_{a,b}} share a vague similarity with q_{\phi_{a}}. However, sampling from that confused latent space via p_{\theta_{a}} and p_{\theta_{b}} results in bad reconstruction.
> > >
> > > On the other hand, the tVAE finds a more congruent latent space projection via q_{\phi_{a}}, q_{\phi_{b}}, and q_{\phi_{a,b}}. But it also shows no clear seperation between labels (green, violet, and pink are strongly confused).
> > >
> > > Finally, our model finds the most coherent and congruent latent space projection of AB by its' encoders q_{\phi_{a}}, q_{\phi_{b}}, and q_{\phi_{a,b}}. Furthermore, they show the clearest separability and best reconstructability.
> > >
> > > ----------------------------------------------------------------------------------------------------------------------------------------
> > >
> > > ----------------- Comment by the reviewer -----------------
> > > "The model learned by tVAE is derived from the JMLL. q(z|a) and q(z|b) are learned separately from the model."
> > >
> > > ----------------- Answer by the authors -----------------
> > > That is correct. But the separate learning of q(z|a) and q(z|b) is what we consider as a "not fully analytical derivation". The training of q(z|a) and q(z|b) has no impact on shaping the latent space and also not on the joint encoder q(z|a,b).
> > >
> > > ----------------------------------------------------------------------------------------------------------------------------------------
> > >
> > > ----------------- Comment by the reviewer -----------------
> > > "Didn't you say that tVAE and JMVAE-Zero were not derived from the JMLL, yet you compared to them? So you could also compare to JMVAE-kl given how similar it is to your model. Right?"
> > >
> > > ----------------- Answer by the authors -----------------
> > > We apologize for confusing the reviewer with too implicit information. We consider the M²VAE as "fully derived" from the JMLL while tVAE and JMVAE-Zero are just "derived" utilizing the JMLL as a starting point.
> > >
> > > However, you are right with the fact that the JMVAE-kl is indeed similar to ours it is, therefore, worth comparing it against the other approaches. We will include the evaluations in our future publications but also want to forecast that the missing uni-modal reconstruction terms in the loss function result in noticeable confusion if uni-modal observations are incomplete (as in our MoG example).

---

### Official Review · AnonReviewer1 · 2018-11-02
**interesting  topic; potential technical error.**

**Rating:** 4
**Confidence:** 3

**Review:**

The paper proposes a multi-modal VAE with a variational bound derived from chain rule.

Pros:
It is an interesting and important research direction.
The presentation is in general clear.

Cons:
1. The re-visit of JMVAE seems not precise. The JMVAE should bound the joint p(a, b) not log p(a|b)p(b|a).
2. Due to the potential misunderstanding of JMVAE, the paper uses the JMVAE bound for log p(a|b) + log p(b|a) in equation (5), which seems wrong.
Equation (4) &(5) itself seems confusing alone. It says L_m = log p(a,b) in (4) then L_m = log p(a|b) + log p(b|a) in (5).
3. If I am not mistaken the error above, the proposed bound is in fact wrong.
4. Assume that the method is correct, with a massive amount of beta:s, I doubt the method would be very sensitive to beta tuning. The experiments just presented some examples of different betas. Quantitive evaluation of beta and performance is needed.
5. To generate multi-modal data, other methods such as VAE-CCA or JMVAE are able to that as well. It is not unique to the proposed method.
6. The experiments are very toyish. The multi-modal data were generated. The method should be evaluated with a real-world benchmarking multi-modal dataset.

---

> ### Author Response · Authors · 2018-11-08
> **Reply to reviewers cons**
>
> We thank the reviewer for his/her work. We hope our responses below may help clarify the scope of our work and its significance.
>
> 1. The original JMVAE proposed by Suzuki et al. [1] does, in fact, optimize the bound of the VI log(p(a|b))+log(p(b|a)) as derived in the appendix A of their work [1]. Further, they derive the ELBO for the VI by substituting the reconstruction terms by the ELBO of the joint multi-modal VAE.
>
> 2. Given 1., we can substitute log p(a|b) + log p(b|a) with L_m. But in fact, there is a typo in eq. (4) & (5) which definitely causes confusion. The left-hand-side (LHS) term should be L_m², denoting the proposed VAE, and not L_m (which denotes the VAE by Suzuki et al.).
>
> 3. As explained in 2. and by correcting the typo in the LHS term, the bounds are correct. We apologize for this confusion and upload a corrected version of the paper. (Origin of the issue: we used the superscript symbol 2 which was dropped by the LaTeX compiler)
>
> 4. In fact, beta tuning is pretty robust as shown by Higgins et al. [2]. One only has to care about the proper ratios between the size of the observable and latent dimensions as briefly explained by us: “... our main concern is the balance between the input (D) and the latent space (Dz) using a constant normalized factor βnorm=βD/Dz.”
>
> 5. As we do not know if this statement relates to our proposed M²VAE (section 3) or the multi-modal data generation via the CVAE (section 4.1), we try to answer this concern two-fold: 5.a relates to the approach of our proposed M²VAE model while 5.b relates to the correlated data generation via the CVAE.
>
> 5.a. From our point of view, current VAE approaches do not explicitly care or investigate the coherence of the latent space in multi-modal cases. Thus, the encoder networks for example in a bi-modal case, i.e. q_\phi_a(z_a|a), q_\phi_b(z_b|b), and q_\phi_{a,b}(z_{a,b}|a,b), may project the observations into different latent spaces (z_a != z_b != z_{a,b}). This happens for the tVAE and JMVAE-Zero, as qualitatively shown in our results, due to the fact that these approaches do not maintain a fully analytical way for deriving the VAE’s objective from the joint marginal log-likelihood (JMLL). In our work, the VAE’s objective has been consequently derived from the JMLL which also disproves statements in related papers, which state that this approach is not applicable. As a result, our model shows coherence in the latent spaces (z_a = z_b = z_{a,b}) regarding the sampled latent space as well as the ELBO for each sample which is, from our point of view, a significant and unique finding in the field of multi-modal generative models. Furthermore, the trained latent space should serve as a much better representation, i.e. feature extractor for subsequent models, which is out of scope for this contribution.
>
> 5.b. We found that current available multi-modal datasets do not satisfy our needs to show the significance of our M²VAE approach. As stated in our contribution, multi-modal data for e.g. sensor fusion should show a correlation. Available datasets [3] may hold this requirement, but may be too complex in nature to qualitatively prove the distinction between SOTA VAEs and our proposed M²VAE. Therefore, we proposed a technique to correlate different datasets (MNITS and fashion-MNIST in our case) by superimposing the latent space of distinctively trained CVAEs. The simultaneous sampling from the various CVAE decoders then produces a correlated dataset, that allows us to show the benefits of our approach.
> If one would just feed for instance MNIST and f-MNIST into a multi-modal VAE, even by keeping the labels the same, the VAE would train an orthogonal distribution per label. This shows that if one feeds uncorrelated data into any VAE, the VAE will obtain the uncorrelatedness as shown in Figure 2 Bottom-Left. One can see that per label, almost all distributions remain orthogonal to each other, which proves our assumption. However, sampling from that latent space will again result in uncorrelated data.
> On the other hand, a multi-modal VAE is able to obtain the correlation of the proposed sampled dataset e-MNIST, which suites our needs for further evaluations in this contribution.
> However, we would be very thankful if the reviewer could point out the literature to the named models (VAE-CCA or JMVAE) which investigate similar nature in VAEs.
>
> 6. As stated in 5.a and 5.b our main objective is to show that our model learns a correct coherent latent space between modalities. However, we would be again very thankful if the reviewer could point out datasets which satisfy the needs for sensor-fusion as well as the necessary complexity to improve the visibility of our work.
>
> [1] Suzuki, M., et al.  (2017). Joint multimodal learning with deep generative models
> [2] Higgins, I., et al. (2016). Early Visual Concept Learning with Unsupervised Deep Learning.
> [3] https://en.wikipedia.org/wiki/List_of_datasets_for_machine_learning_research

---

### Official Review · AnonReviewer3 · 2018-11-06
**Disjointed paper on an interesting topic**

**Rating:** 4
**Confidence:** 2

**Review:**

This paper introduces a new VAE model (JMVAE) for multi-modal data with a
shared latent representation. An method is also introduced to synthetically
created bi-modal datasets with correlated latent representations.

The writing was a little awkward to follow at times, and I'm still not
sure what Ι am suppose to take away from the figures plotting the latent
representation. The evaluation is fairly qualitative and it's difficult to
understand what we achieving from using JMVAE.

I'm not clear what the contribution of this work provides, as there is already
plenty done on learning multi-modal representations.

One weakness with this work is all the examples are fairly toy
problems. The article motivates the work as combining raw multi-modal
sensor datasets, but no real tasks are shown.

---

> ### Author Response · Authors · 2018-11-08
> **Explaining takeaway message and applicability of toyish dataset**
>
> We’d like to thank the reviewer for their thorough review and helpful suggestions. We will improve our work further but also want to start a discussion to clarify the significance of our contribution.
>
> From our point of view, current multi-modal VAE approaches do not explicitly care or investigate the coherence of the latent space in multi-modal cases. Therefore, exploiting multi-modal VAEs as e.g. feature extractor may be questionable by means of the manifold theory. It happens for SOTA VAEs, that the encoder networks for example in a bi-modal case, i.e. q_\phi_a(z_a|a), q_\phi_b(z_b|b), and q_\phi_{a,b}(z_{a,b}|a,b), may project the observations into different latent spaces (z_a != z_b != z_{a,b}). However, in our work, the VAE’s objective has been consequently derived from the joint marginal log-likelihood which also disproves statements in the related papers, which state that this approach is not applicable. As a result, our model shows coherence in the latent spaces (z_a = z_b = z_{a,b}) regarding the sampled latent space as well as the ELBO for each sample which is, from our point of view, a significant and unique finding in the field of multi-modal generative models. However, the takeaway message from the figures showing the latent representations is as follows: If the latent spaces (z_a, z_b, z_{a,b}) found by the various encoders (q_\phi_a(z_a|a), q_\phi_b(z_b|b), and q_\phi_{a,b}(z_{a,b}|a,b)) are the same, z_a and z_b should be sub-spaces of z_{a,b}. This should result in a congruence of samples and in a similarity of the ELBO per sample between the subspaces. Furthermore, the trained latent space should serve as a much better representation, i.e. feature extractor for subsequent models, which is out of scope for this contribution.
>
> To explain the application of the “toyish” dataset, our main objective is not to show that our model performs best on complex multi-modal data, but to show the unique property of our proposed model that it learns a correct coherent latent space between modalities. We found that current available multi-modal datasets do not satisfy our needs to show the significance of our M²VAE approach. As stated in our contribution, multi-modal data for e.g. sensor fusion should show a correlation where it is fuseable. Available datasets, i.e. as listed in [1], may hold this requirement, but may be too complex in nature to qualitatively prove the distinction between SOTA VAEs and our proposed M²VAE. Therefore, we proposed a technique to correlate different datasets (MNITS and fashion-MNIST in our case) by superimposing the latent space of distinctively trained CVAEs. The simultaneous sampling from the various CVAE decoders then produces a correlated dataset, that allows us to show the benefits of our M²VAE approach.
>
> However, we would be again very thankful if the reviewer could point out non-toyish datasets which satisfy the needs for sensor-fusion as well as the necessary complexity to improve the visibility of our work.
>
> [1] Suzuki, M., Nakayama, K., & Matsuo, Y. (2017). Joint multimodal learning with deep generative models, 1–12. Retrieved from https://arxiv.org/abs/1611.01891

---

### Meta-Review · Area_Chair1 · 2018-12-09
**Poorly motivated, implications unclear**

**Confidence:** 4
**Recommendation:** Reject

**Metareview:**

This paper suggests a problem with the standard ELBO for the multi-modal case, and proposes a new objective to address this problem.  However, I (and some of the reviewers) disagree with the motivation.  First of all, there's no reason one can't train a separate encoder for every combination of modalities available, at least when there are only 2 or 3.  Failing that, one could simple optimize per-example approximate posteriors without using an encoder.

Second, once you stop optimizing the ELBO, you've lost the motivating principle for training VAEs, and must justify your new objective empirically.  Almost all of the results are (in my opinion) ambiguous plots of latent encodings.

Finally, a point made throughout the paper and discussions was that different modalities should give the same encodings, which is plainly false.  One of the reviewers made this point: "The fact that z_a != z_b != z_{a,b} should be expected if a and b provide different information. I don't see the problem with this.", which you dismiss.  Additionally, the encoder's job is to approximate the true posterior.  The true posteriors will in general be different for different modalities.

I would recommend focusing on ways to train the original ELBO in the presence of different modalities, instead of modifying it based on these intuitions.